# Assessment of the Risk of Contamination of Food for Infants and Toddlers

**DOI:** 10.3390/nu13072358

**Published:** 2021-07-09

**Authors:** Anita Mielech, Anna Puścion-Jakubik, Katarzyna Socha

**Affiliations:** Department of Bromatology, Faculty of Pharmacy with the Division of Laboratory Medicine, Medical University of Białystok, Mickiewicza 2D Street, 15-222 Białystok, Poland; anita.mielech@umb.edu.pl (A.M.); katarzyna.socha@umb.edu.pl (K.S.)

**Keywords:** baby food, food contaminant, food safety

## Abstract

Infants and toddlers are highly sensitive to contaminants in food. Chronic exposure can lead to developmental delays, disorders of the nervous, urinary and immune systems, and to cardiovascular disease. A literature review was conducted mainly in PubMed, Google Scholar and Scopus databases, and took into consideration papers published from October 2020 to March 2021. We focused on contaminant content, intake estimates, and exposure to contaminants most commonly found in foods consumed by infants and children aged 0.5–3 years. In the review, we included 83 publications with full access. Contaminants that pose a high health risk are toxic elements, acrylamide, bisphenol, and pesticide residues. Minor pollutants include: dioxins, mycotoxins, nitrates and nitrites, and polycyclic aromatic hydrocarbons. In order to reduce the negative health effects of food contamination, it seems reasonable to educate parents to limit foods that are potentially dangerous for infants and young children. An appropriate varied diet, selected cooking techniques, and proper food preparation can increase the likelihood that the foods children consume are safe for their health. It is necessary to monitor food contamination, adhere to high standards at every stage of production, and improve the quality of food for children.

## 1. Introduction

Infants and young children are particularly vulnerable to contaminants in food because of the physiological characteristics that distinguish them from adults. Exposure to potentially toxic substances is especially dangerous because of infants’ and young children’s higher food intake (per kilogram of body weight—kg/bw), higher ventilation (kg/bw), and greater body surface area (kg/bw). Infants and toddlers have higher resting metabolic rates, which also contributes to greater sensitivity to toxins [1].

Non-food factors can also be a source of some contaminants such as toxic elements, which in the case of lead (Pb) can account for up to 61% of the total intake. Most gastrointestinal functions develop by the first year of life, but intestinal motility remains slow, intestinal transit time is prolonged and the small intestine is incompletely developed, therefore absorption of toxic elements is higher compared to adults. The urinary and biliary systems also mature in the first year of life, while the mechanisms responsible for filtering and elimination of chemicals are not fully developed. In young children, the rate of gastric emptying is higher than in adults, resulting in more rapid absorption and higher peak serum concentrations [2]. Infants and young children are particularly susceptible to food contamination due to the high sensitivity of their digestive tract, and therefore intake limits for toxic substances should not be exceeded. Characteristics of contamination levels with selected compounds that are safe for infants and young children are presented in Table 1.

Food for children should be a source of nutrients, vitamins, and minerals. It should also be free from contaminants that may adversely affect their health and development. Therefore, the aim of the study was to review the literature on selected contaminants in food intended for infants and young children and to assess the risk of its consumption.

## 2. Materials and Methods

This publication reviews research, published from 2004 to 2021, on risk assessment of food contaminants in infant and young child products. We focused on analysis of studies that considered such aspects as contaminant content, estimation of intake, and exposure to contaminants most frequently found in food consumed by children aged 0.5–3 years. We made a preliminary selection by searching the PubMed database and Google Scholar under the headings ‘food safety’, ‘food contamination’, and ‘environmental contaminants’. On this basis, the most common contaminants in baby food were selected for inclusion in this review.

The literature review was mainly conducted in the PubMed database, as well as Google Scholar and Scopus, from October 2020 to March 2021. Studies with the largest numbers of trials were selected for consideration. The database was searched for ‘contamination’, ‘baby food’, ‘infant formula’, ‘toxic elements’, ‘dioxins’, ‘acrylamide’, ‘bisphenol’, ‘furan’, ‘mycotoxins’, ‘nitrates’, ‘polycyclic aromatic hydrocarbon’, ‘pesticides’, ‘3-MCPD’, ‘glycidyl esters’, and ‘mineral oil hydrocarbons’. The full texts of the selected papers were analyzed to assess whether they met the accepted criteria. Publications available in a language other than English were excluded. 

The reference list of each article was searched to ensure that no important research articles were missed. Included clinical trials provide the latest evidence on food safety for infants and toddlers. No systematic review was undertaken, because this topic involves too many different groups of food contaminants.

This review discarded studies that involved animal measurements of contaminant concentrations, prenatal exposure, adult exposure to dietary contaminants, and contaminant-exposed sick children. Research that analyzed conventional and infant foods as one group or estimated the risk of exposure to contaminants on the basis of a dietary record were also excluded. Other reasons for exclusion were: age group (under 6 months, over 3 years old), insufficient sample/study size, lack of full access, the study being a literature review, failure to specify products for children, inappropriate study material (e.g., human milk). Studies that did not serve the purpose of this review were also rejected. 

## 3. Results

This paper presents the most important publications describing the current state of knowledge on risk assessment of contaminants in food for infants and toddlers.

This review summarizes scientific evidence from the past 17 years. A total of 6278 studies were identified, 2880 duplicates were removed, 4298 related titles were chosen. Of these, full access was given to 146 publications, but 63 articles met the exclusion criteria, and 83 studies were eventually included in this review. More information about the search and selection process of studies is shown in Figure 1—based on Preferred Reporting Items for Systematic Reviews and Meta-Analyses (PRISMA) guidelines, with own modifications.

### 3.1. Toxic Elements

A study by Igweze et al. (2020) found that 96% of infant formulas exceeded the daily allowable concentration of Pb (0.4 mg/kg) and 53% exceeded the maximum Cd (cadmium) concentration (0.1 mg/kg), as per the Food and Drug Administration (FDA) standards [9]. Another study also reported Pb contamination in 37% of samples and Cd contamination in 57% of infant food samples. There was no significant difference in toxic element content between organic and conventional cultivation systems. Rice-based products contained significantly more Pb and Cd [10]. Some authors only detected Pb in individual samples [11,12]. In a Brazilian study evaluating Pb and Cd concentrations in infant formulas (formulas for particular nutritional uses for infants and breast milk substitutes), the median Pb concentration was 0.109 mg/kg, exceeding the maximum standard set by the Food and Agriculture Organization of the United Nations/World Health Organization (FAO/WHO) (0.02 mg/kg). In addition, Pb concentrations in 7 infant formulas and 22 milk samples exceeded the Brazilian standard (0.2 mg/kg in infant formula and 0.05 mg/kg in milk) [13]. 

In another study, Cd was detected in 65% of the samples, of which the highest amounts were found in potatoes, cookie cereal bars, vegetables, and bread. Additionally, 2.5% of children aged 3 years and under exceeded the TWI for Cd [14]. 

In contrast, a study by Gao et al. (2020) found no exceedance of Cd intake standards in snacks consumed by children aged 1-6 years. The estimated daily intake did not exceed the tolerable daily intake (maximum tolerable daily intake (MTDI) = 46 mg/d). Among the studied snacks, the most Cd was observed in flour products (37% of total Cd delivery from snacks) [15]. 

In a study analyzing Hg (mercury) content in 291 children’s products, none of the samples exceeded the maximum allowable concentration. Hg was detected in prepared fish- and meat-based products (maximum concentration of 7.4 µg/kg), in dairy desserts, cereal products and in one infant formula [16]. In contrast, in another study, fish products exceeded the maximum residue limits (MRL) for Hg (0.5–1.0 mg/kg) in as many as 66% of all samples [17]. 

A study evaluating long-term exposure of children under 6 years to toxic elements reported that Pb exposure exceeded the reference value (RfT = 0.5 µg/kg bw) among 35% of children, 42% of children exceeded the TWI of Cd, but Hg exposure was below the TWI. The main sources of Pb exposure were dairy products and vegetables, while in the case of Cd and Hg, it was cereal products, fish, and shellfish [18]. Two of the analyzed studies also contain evidence to suggest that infant formulas are safe in terms of Pb content [19,20].

A study evaluating arsenic (As) in infant foods found that 75% of samples were contaminated with inorganic As. The highest concentrations of As were found in rice noodles, whole grain rice, and crackers [21]. High As contamination was also reported by Rotenberg et al. (2017) and Ljung (2011) [22,23]. 

Another study evaluated the contents of As, Cd, Hg, and Pb in infant foods. None of the products contained amounts of As which would pose a health risk. Oatmeal had the highest concentration of Cd (1.02 µg/g), and fish had the highest concentrations of As and Hg (1.02 µg/g, 6 µg/g, respectively) [24]. 

A summary of the studies on the concentration of toxic elements is shown in Table 2.

### 3.2. Acrylamide

Abt et al. (2019) evaluated infant dietary exposure to acrylamide. Products intended for infants up to 12 months (e.g., snacks) had higher levels of acrylamide than jarred infant foods, suggesting the need to intensify acrylamide reduction processes during food production. The highest concentrations of acrylamide were observed in baby potato crisps, crackers, and breakfast cereals, i.e., complementary foods for infants and toddlers. The mean estimated intake of acrylamide for ages up to 2 years was 0.42 µg/kg bw/day [25]. A large sample size study (n = 2517) showed that the estimated dietary intake (EDI = 2 µg/kg body weight/day) was exceeded among 7% of children aged 12–36 months. Acrylamide was identified as a potential hazard so reduction of this substance during processing is of utmost importance (Margin of Exposure − MOE < 10,000) [26]. 

In a study by Lambert et al. (2018), acrylamide was detected in 113 samples (80%) and its mean concentrations varied from 0.14 to 102.00 µg/kg. The highest value of acrylamide was found in infant cookies (99.5 µg/kg), potatoes with carrots (67.0 µg/kg), and potatoes with pumpkin (67.0 µg/kg) [27]. Elias et al. (2017) detected the highest concentrations of the substance in potato and cereal snacks. Above-limit concentrations of acrylamide were found primarily in plant-based infant foods. The mean value of acrylamide was 30–65 μg/kg. High amounts of acrylamide in infant products entail serious health risks (MOE < 10,000) [28]. In a study by Mojska et al. (2021), the average exposure of infants aged 6–12 months to this contaminant ranged from 2.1 to 4.3 μg/kg bw/day, which exceeds the reference values. In addition, it was reported that the main sources of acrylamide were jarred baby meals (56.7%) and follow-on infant formulas [29]. 

A summary of the studies on the concentration of acrylamide is presented in Table 3.

### 3.3. Bisphenol

In a study by Cao et al. (2009), bisphenol content was determined in 122 infant products packaged in jars and metal lids. Bisphenol was detected in 99 products, but its mean concentrations were mostly low (0.54–1.1 ng/g) [30]. In another study, bisphenol was detected in 38% of infant formula samples (mean concentration: 0.26 µg/L) and in 76% of breast milk samples (mean concentration: 1.3 µg/L) [31]. In Cirillo et al. (2015), bisphenol was found in 60% of baby foods and its mean concentration was 0.021 µg/g. Both liquid and powdered products had similar bisphenol contents, suggesting that the substance came from manufacturing processes rather than food packaging [32]. In a study by Niu et al. (2007), bisphenol was detected in 36% of samples. Mean bisphenol exposure among infants up to 12 months of age exceeded the tolerable daily intake (TDI, 5–17 μg/kg bw/day). No exceedances of the TDI were reported among adults, confirming the higher risk of exposure among children [33]. Karsauliyta et al. (2021) studied the concentration of bisphenol analogues in infant formulas for 0- to 12-month-olds, in which they found exceedances to bisphenol [34]. The opposite result was obtained by Sun et al. (2017), who did not detect bisphenol in any of the analyzed 76 baby food samples [35]. 

A summary of the studies on the concentration of bisphenol A can be found in Table 4.

### 3.4. Dioxins

In a study by Saito et al. (2008), the dioxin content of infant food was evaluated. It was observed that the intake of dioxins in food was safe [36]. Lorán et al. (2010) concluded that dioxins in baby foods (cereal products, meat dishes, fish dishes) did not pose a health risk to infants [37]. Sasamoto et al. (2006) assessed that the main sources of dioxin were ready-made infant products and breast milk. The intake in all study groups (infants and young children from 5 to 15 months of age) was lower than the TDI (4 pg/kg/day) [38]. Similar observations were noted in other studies [39,40]. In Hulin et al. (2020), the dioxin content of traditional foods was higher than that of infant foods. In addition, the TDI was exceeded among children older than 6 months of age (exceedance of the TDI was 4.5% among children of 7–12 months and 5.1 to 7.4% among children of 13–36 months of age). It was estimated that approximately 96% of the food consumed by children contributed to dioxin exposure [41]. Research published by authors from Italy showed that children could safely consume the products under investigation [42].

A summary of the studies on the concentration of dioxins is shown in Table 5.

### 3.5. Furan

A study by Lambert et al. (2018) assessed furan content in samples of food for infants and children under 3 years of age. Furan was detected in 113 samples, with the highest amounts in spinach (95 µg/kg,) carrots, and ham (73 µg/kg) [43]. In another study, it was observed that heating in a microwave oven reduced furan content to 35%, while using a water bath reduced it to 53%. The highest amounts of furan were found in meat-based products (7.9–61.0 ng/g) and fish (19.0–84.0 ng/g) [44]. A significantly higher estimated daily intake of furan by infants (0.333 μg/kg) than by adults (0.093 μg/kg) was also observed. This is particularly worrisome since ready-made infant products are often a staple food during this period of life [45]. Similar results were obtained by Scholl et al. (2013), who also showed that infants had a significantly higher risk of furan exposure than adults [46]. In another paper, it was shown that the estimated dietary intake of furan among infants ranged from 0.03 to 3.56 μg/kg and exceeded the RfD reference standard (0.1 μg/kg) [47]. Liu demonstrated that the average concentration of furan in infant formulas ranged from 2.4 to 28.7 ng/g, making infant products potentially hazardous to children’s health [48]. 

Lachenmeier et al. (2009) estimated that the mean furan exposure among infants was 0.2 µg per kg body weight a day, which is not a health risk. In addition, it was shown that in contrast to jarred infant foods, none of the home-prepared foods contained furan [49]. 

A summary of the studies on the concentration of furan can be found in Table 6.

### 3.6. Mycotoxins

In a study by Mallmann et al. (2020), mycotoxins were detected in 31% of cereal product samples and in 19% of infant cereal porridge samples. In addition, co-occurrence of 2 or more mycotoxins was found in 31% of cereals and 19% of infant cereals. The most frequently detected mycotoxins were fumonisins (26.7%) and zearalenone (14.8%) [50]. Saleh et al. (2019) evaluated patulin content in apple-based products for children. Estimated daily intake did not exceed the maximum TDI, i.e., 0.4 μg/kg bw/day. The most exposed group were children under 6 years of age [51]. In contrast, another study detected aflatoxins in 20% of the samples (aflatoxin B1), of which 10% exceeded the maximum tolerable concentration (0.1 µg/kg). There were no differences between whole grain products and refined cereal products, while organic cereal products contained higher concentrations of deoxynivalenol than conventional products [52]. In a study by Postupolski et al. (2019), none of the 302 samples of cereal products for children exceeded the TDI. For medium exposure, the values were up to 3%, while for high exposure, up to 10% of the reference values (mainly deoxynivalenol and fumonisins) [53]. In another study, ochratoxins were found in 41% of cereal products for children, the average level being 0.42 ± 0.27 μg/kg. It was observed that 7.8% of the samples exceeded the highest permissible ochratoxin concentration recommended by the European Commission (0.5 μg/kg). The highest ochratoxin contamination was found in rice-based infant products (57%), wheat (23%), and multi-grain products (20%) [54]. In another study, aflatoxin and fumonisin contamination concerned 42% of all cereal- and nut-based products. Mycotoxin exposure exceeded reference values, indicating a health risk to infants and young children [55]. In another study, 85% of the analyzed infant formula samples exceeded the maximum tolerance limit (MTL) [56]. In Sundheimet al. (2017), the highest concentration of deoxynvalenol (DON) mycotoxin was observed among children aged 2 years, which was associated with high consumption of cereal products. The average exposure was twice the TDI [57].

There are also studies which indicate that the risk from mycotoxins in infant formulas is negligible [58,59,60]. In conclusion, contamination is rare and cereal products for infants and young children appear to be safe. 

A summary of the studies on the concentration of mycotoxins is presented in Table 7.

### 3.7. Nitrates and Nitrites

A study by Vasco et al. (2011) analyzed the nitrate content of foods for children (processed vegetables, fruit, juices) from organic and conventional farming. Only one sample exceeded the acceptable daily intake. Additionally, no differences were noted between nitrate concentrations in foods obtained using different farming methods [61]. In another large study (n = 1150), none of the infant products analyzed exceeded the maximum acceptable nitrate level. The estimated daily intake of nitrates by infants and children over 1 year of age was 13% and 18% of the ADI, respectively [62].

On the other hand, in a study by Cortesi et al. (2014), no exceedance of the maximum allowable nitrate concentration (200 mg/kg bw) was recorded in the analyzed samples [63]. Similar observations were also made by other researchers [64]. Another study reported single exceedances of the maximum allowable nitrate concentration in vegetable preparations for this age group, with the mean nitrate concentration equal to the upper limit of the standard (189 mg/kg bw) [65]. In a study by Elias et al. (2020), where the source of nitrates and nitrites were the most common processed meat products, the exceedance of ADI was reported in 3% of children [66]. Mancini et al. (2014) showed significant exceedances of the ADI of nitrite content in food for children of 7–12 months and 13–36 months by 16% and 58%, respectively. Meat was the main source of high nitrite concentrations. Considering the whole study population, the average intake of nitrites was lower than the ADI, confirming the fact that infants and children are the group that is most exposed to nitrites in food [67]. Similar observations were noted for a smaller group of subjects [68].

A summary of the studies on the concentration of nitrates and nitrites is shown in Table 8.

### 3.8. Pesticide Residues

A study by Nougadere et al. (2020) evaluated pesticide content in foods for children under 3 years of age and in commercial foods. Pesticide residues were detected in 67% of samples (78 different pesticides). Among the most frequently detected were fungicides, 2-phenylphenol, boscalid, azoxystrobin, captan, and tetrahydrophthalimide [69]. Similar results were obtained by Stepan et al. (2005), who found pesticides in 60% of all analyzed samples. In infant fruit foods, pesticide residues were found in 16% of samples, with maximum residue levels (0.01 mg/kg) exceeded in 9% of them. The most common pesticides were organophosphorus insecticides and fungicides, represented by phtalimides, dicarboximides, and sulphamides [70]. In a study by Jeong et al. (2014), at least 1 pesticide was detected in every sample. It was reported that the average concentration of organochlorine pesticides was 2 times higher in foods for 15-month-olds than in products for 6-month-olds [71]. In a study by Torović et al. (2020), pesticides were detected in 56% of infant food samples. In addition, more domestic products (85%) than imported products (38%) had high pesticide contents [72]. Kapoor et al. (2012) evaluated the content of imidacloprid, an alternative to organochlorine pesticides. Imidacloprid was found in 15% of the samples, of which 3% exceeded the MRL [73]. Gilbert-Lòpez et al. (2007) assessed the content of 12 pesticides in baby food containing fruit. The detection limit of the developed method ranged from 0.1 (imazalil) to 4 µg/kg (iprodione). Despite the fact that three pesticides (carbendazim, imazalil, and thianendazole) were detected in approximately 60% of the samples, none exceeded the limit of 0.01 mg/kg [74]. One more recent study, published by Panseri et al. (2020), found perchlorate in 10.5% of the baby food and commercial food samples the authors tested [75].

Table 9 summarizes the studies on pesticides in baby food.

### 3.9. Polycyclic Aromatic Hydrocarbon (PAH)

Dairy products showed higher PAH concentrations than meat- and fish-based products. In low-fat products (up to 3% fat), PAH concentrations were lower (19.4 µg/kg) than in products with higher fat content (43.3 µg/kg). PAH norms were exceeded in milk based products in 18.2% (benzopyrene) and 77.7% (ΣPAH4) of samples and also in meat- and fish-based products (5.6% and 44.4% respectively) [76]. Similar observations were made by Di Bella et al. (2020), where exceedances were reported in the PAH content of dairy products: cow milk (mean content 12.56 ng/g), sheep and goat milk (9.2 ng/g), with a maximum concentration permitted for infants and young children of 1 ng/g. Concentrations of aromatic hydrocarbons in meat and fish products did not exceed maximum allowable levels [77]. In contrast, another study reported that children’s intake levels of PAHs in conventional products were safe. The average PAH intake was 192 ng/day, while the maximum daily intake was 1575 ng/day [78]. A study by Badibostan et al. (2019) evaluated exposure to polycyclic aromatic hydrocarbons in infant and toddler formulas. The authors found that benzopyrene was present in 64.3% of samples, chrysene in three samples, and fluoranthene in one sample. One product had a concentration of 1.43 μg/kg and that sample exceeded the maximum tolerable limit MTL (1 μg/kg) [79]. On the other hand, some of the studies under review deemed baby products to be safe as regards PAH [80,81]. 

A summary of the studies on the concentration of PAH is included in Table 10.

### 3.10. 3-Monochloropropane-1,2-Diol (3-MCPD) and Glycidyl Esters

Beekman et al. (2020) found higher concentrations of 3-MCPD than those of glycidyl esters in 23 samples (10%). Decreased concentrations of 3-MCPD (about seven-fold lower) and glycidyl esters (about three-fold lower) were noted during a 3-year follow-up, which may be due to the use of new technologies that reduce the risk of contamination. Additionally, higher concentrations of 3-MCPD and glycidyl esters were observed in powdered products than in ready-to-eat ones. Glycidyl is found in vegetable oils, but the highest amounts are found in palm oil [82]. Another study also confirmed lower concentrations of 3-MCPD and glycidyl esters compared to previous years [83]. In a study by Campi et al. (2020), the highest concentrations of 3-MCPD and glycidyl were observed in seed oil, margarine, and cookies. It was also reported that palm oil increased the concentration of MCPD and glycidyl esters in products [84].

Over 71% of diet and dairy product samples were contaminated with 3-MCPD and glycidyl esters. Estimated daily exposures to bound 3-MCPD (0.48–0.49 μg kg/bw) and glycidyl esters (1.00–1.11 μg/kg/bw/day) did not exceed the limits and did not constitute a health risk to children [85]. In another study, the TDI for 3-MCPD was exceeded in a group of children’s snacks (mainly potato crisps, crackers, peanuts, and muesli) and the highest concentration was found in biscuits [86]. Interestingly, the authors from the US showed the lowest concentrations of 3-MCPD in products that contained palm oil [87].

A summary of the studies on the concentration of 3-MCPD and glycidyl esters is given in Table 11.

### 3.11. Mineral Oil Hydrocarbons (MOHs)

In a study by Sui et al. (2020), mineral oil hydrocarbons (MOHs) were detected in 17 of 61 samples, with the highest amounts reported in a goat milk-based infant formula [88]. In Zhang’s (2019) study, MOHs were found in 66% of samples, mostly infant formulas [89]. In another study, MOHs were present in all analyzed samples that contained meat and fish. The highest concentration (2 mg/kg/bw) was recorded in a product containing salmon: it exceeded the MOH standard (maximum limit of 0.6 mg/kg/bw) [8]. Lei et al. (2019) observed the highest risk of MOH exposure among infants aged 0–6 months and 6–12 months. Additionally, MOH intake was higher among Europeans and lower among the Chinese, so food contamination with mineral oil hydrocarbons may constitute a health risk for infants from Europe [90].

Table 12 summarizes the studies of mineral oil hydrocarbons in baby food.

## 4. Discussion

Infants and young children are much more vulnerable to contaminants from food. When planning children’s diets, it is important to pay attention not only to the quantity but also to the quality of food. Unfortunately, products that are potential sources of contaminants are often also desirable in a healthy diet. Contaminants in foods for infants and toddlers can cause adverse health effects, both in the short term and adulthood.

Exposure to Pb in infancy disrupts children’s development by damaging the nervous system. Long-term exposure to this element, even below relatively safe doses, contributes to impaired concentration and attention, possibly affecting the intelligence quotient (IQ) [91]. Any amount of Pb in the body disrupts the nervous system. It has been shown that the higher the concentration of Pb in the blood, the more nervous system disorders develop in children [92]. High amounts of Pb can be found in meat, fish and seafood, grain products, vegetables, fruit, and dairy products [93]. 

Cadmium (Cd) features in the International Agency of Research on Cancer (IARC) list of carcinogenic contaminants. Cd damages the renal tubules, contributing to the dysfunction of the excretory system. In addition, it disturbs the acid-base balance and causes endocrine disruption. As in the case of Pb, there are suggestions that doses lower than the TWI (Tolerable Weekly Intake) may also have a neurotoxic effect in children [14]. The main sources of Cd in foods are grain products, rice-based products, vegetables, fish, and seafood [93]. Mercury (Hg) exposure among infants and young children has a neurotoxic impact through the deleterious effects of methylmercury. Short-term exposure to Hg may predispose individuals to cardiovascular disorders and contribute to immune system dysfunction. The main sources of Hg in food are seafood and fish, the highest amounts found in predatory fish [94]. Inorganic As has been classified as a carcinogen. Exposure to As causes skin diseases, nervous system diseases, neurodevelopmental disorders in children, and lung diseases. Long-term exposure can contribute to cardiovascular disease, type 2 diabetes, and cancerous processes. Dietary sources of As exposure include dairy products, rice, and foods for infants and young children [95]. As far as toxic elements in children’s diets are concerned, special attention should be paid to products potentially contaminated with Pb and Cd, because they carry the greatest health risk. It is worth limiting the consumption of rice and rice products, as they are often contaminated with inorganic As. To exceed the maximum intake of As, a 3-year-old child must eat 242 g of rice cereal per day (6 portions of cereal) or 167 g of rice waffles, which is about 16.5 pieces [96].

Acrylamide is a carcinogenic compound that has been classified as a probable human carcinogen by the IARC. Acrylamide has potent neurotoxic, carcinogenic and genotoxic effects. It disrupts mitochondrial function, leading to cell apoptosis [97]. It also has a toxic influence on enzymatic mechanisms, hormonal balance, muscle function, and fertility. In the case of children, acrylamide mainly comes from heat-treated potatoes (french fries, potato chips, potato pancakes) [98]. Cereals and potato snacks can also be dangerous to this highly sensitive group, so it seems important that remedial steps are taken to reduce the content of acrylamide in infant products.

Bisphenol (BPA) is a substance which, with prolonged exposure, causes endocrine disruption in the body. It is also linked to heart disease, more frequent heart attacks, and ischemic heart disease. BPA exposure is especially dangerous among infants and young children. According to the European Food Safety Authority (EFSA), the TDI is 50 μg/kg/day, but endocrine disruption has been observed at exposures below the TDI [99]. BPA is another substance that may pose a health risk to infants and young children. Changes are needed at the production level to minimize exposure to this contaminant. To reduce bisphenol exposure in children’s diets, more home-made foods than prepared baby foods must be consumed. Additionally, food manufacturers should implement bisphenol degradation procedures.

Dioxins are a group of organic compounds which, together with polychlorinated dibenzodioxins and polychlorinated biphenyls, constitute dioxin-like compounds. Dioxins are stored throughout the food chain in adipose tissue. The largest source of exposure, accounting for more than 90% of exposure, is food, mainly milk, eggs, and meat [39]. Dioxins can be found in breast milk and modified milk—much higher concentrations are observed in breast milk [100]. They cause endocrine disorders, dysfunctions of the central nervous system, fertility problems, and may also contribute to cancer [101]. Many developmental disorders observed in children, including hypotonia, neurodevelopmental and neurobehavioral disorders, lower IQ, hearing disorders, discoloration and dermatological abnormalities, changes in thyroid hormone levels, can be linked to dioxins [102]. As shown in the literature, commercial foods and infant formulas are safe for children in terms of dioxin contamination, as no average intake exceedances have been reported.

Furan has been classified by the IARC as possibly carcinogenic to humans (Group 2B). It is found in stored products that require heat treatment to produce, e.g., bread and pastries, coffee, jarred/canned products, and baby food. Self-prepared meals do not contain furan. Heating food in a closed system causes furan to store in canned and jarred foods. Because of its high volatility, open-system heated meals contain less furan. Furan causes liver damage, kidney damage, and carcinogenic effects [103]. It may be a cause of poorer health among infants and young children, because its estimated intake exceeded the standards in most studies. Practices such as heating food without a lid can greatly minimize this risk.

Mycotoxins are low molecular weight compounds found in cereals and vegetables and their preparations. The most common mycotoxins include: aflatoxins, ochratoxins, fumonisins, and zearalenone. Infants and young children are the most vulnerable to mycotoxins, because they consume large amounts of cereal products in relation to their body weight. Toxin exposure is associated with carcinogenic, nephrotoxic, neurotoxic, hepatotoxic, hormonal, teratogenic, and immunotoxic effects [103]. Mycotoxins cause developmental delay in infants, impaired immune response, gastrointestinal disorders, and also dysfunction in cognitive development. In older children, developmental delays, cognitive and neurological disorders, and learning difficulties are seen [102]. According to the literature, mycotoxin contamination is rare and therefore should not pose a health risk to children.

Sources of nitrate and nitrite in food include nitrogen fertilizer remains, preservative remains from processed foods, and nitrate naturally occurring in green plants as a nitrogen metabolite. Nitrates have no toxic effects, but can convert to nitrites when exposed to bacteria during storage or during digestive processes in the human body. In addition, nitrite reacts with hemoglobin causing oxidation to methemoglobin, which leads to methemoglobinemia in children. This is a dangerous condition for infants because their stomach pH is higher than that of adults, which further facilitates the transformation of nitrates into nitrites. Infants have low expression of NADPH methemoglobin reductase, so the T_1/2_ (half-life) is longer than in adults [104,105]. Nitrates and nitrites affect cardiovascular and gastrointestinal homeostasis through nitrogen oxygen conversion. Most nitrates are found in vegetables, such as: beets, radishes, green leafy vegetables, and celery [106]. Meat and meat products, as well as vegetables and fruits present in baby products, are safe in terms of nitrates and nitrites.

Pesticides include dangerous agents such as organochlorine pesticides, as well as less harmful substances. Organochlorine compounds are characterized by a high ability to accumulate and a long half-life, which makes them very toxic to the human body [107]. The sources of human exposure to pesticides are meat, fish, dairy products, and drinking water. The use of certain agents, such as the organochlorine pesticide DDT, is restricted or banned in some countries. These toxins can cause acute or chronic health effects, e.g., neurodegenerative diseases, neurological disorders, pro-oncogenic processes, respiratory disorders, or cardiovascular dysfunctions. Exposure of infants and young children to pesticides can cause neuro-behavioral disorders [108]. Pesticide residues in food, present both in animal and plant products, are also a significant danger. The most effective method to reduce the consumption of pesticides in food is to soak vegetables in a solution of baking soda (10 g of baking soda and 1 L of water). Peeling fruits and vegetables, washing in water, refrigeration, blanching, pasteurization, and cooking are also recommended [109,110]. These methods should be applied not only by individual consumers but also by producers as part of good manufacturing practices. It seems appropriate to make it obligatory for manufacturers to control pesticide content in commercial products for children.

Polycyclic aromatic hydrocarbons (PAHs) are pollutants with genotoxic effects, possibly carcinogenic to humans. One of the best studied PAH compounds is benzopyrene, classified as group 1 by the IARC. Benzopirene and Σ4PAH (benzopyrene, chrysene, benzoaanthracene, and benzobfluoranthene) are used to determine PAH content in food products [77]. PAHs are not found in unprocessed food, but can be formed during food processing. Polycyclic aromatic hydrocarbons do not seem to constitute a health threat to infants and young children.

3-Monochloropropanol-1,2-diol (3-MCPD) esters and glycidyl esters are impurities that arise during the refining of cooking oils. Esters undergo hydrolysis processes in the gastrointestinal tract, releasing, among other things, 3-MCPD and glycidyl. 3-MCPD has been recognized as a possible human carcinogen by the IARC. Glycidyl has also been found to be a probable human carcinogen (B1), so it is recommended that its consumption should be as low as possible [82]. 3-MCPD and glycidyl esters may pose a risk, but with new technologies this problem is becoming less of a concern, as their concentrations are decreasing. 

Mineral oil hydrocarbons (MOH) are a product of petroleum transformation. Contamination of food with MOH is caused by contaminants associated with food packaging, or contamination of plants and, consequently, vegetable oils [111]. Mineral oil hydrocarbons may cause a health hazard, so more research is needed.

Geographic variation of crops also influences food contamination levels. The content of toxic elements and pesticides is determined by the use of agrochemicals and the industrial activity of a region [112]. For example, the environment in which rice is grown affects the bioavailability of As found in it. Higher levels of As are reported in Bangladesh, China, Taiwan, Vietnam, and Thailand [113]. In the case of mycotoxins, the highest levels of pollution are recorded in certain regions of Africa and Southeast Asia, because of the conditions of tropical and subtropical climate [114]. In a study by Hossain et al. (2015), mycotoxin contamination was observed in North America and Asia, while samples from Europe were contamination-free [115]. High temperature is a major factor in mycotoxin contamination, therefore higher levels are observed in southern Europe than in northern Europe [116]. 

## 5. Conclusions

Dietary diversity in children may indirectly contribute to reducing the risk of exposure to contaminants. Contaminants that have been shown to be potentially dangerous, such as: toxic elements, acrylamide, bisphenol A, pesticides, and MOH, need to be closely monitored.

Children’s meals should be as varied as possible, not based on only one food group (e.g., rice) in order not to exceed the doses of toxic elements. To minimize the risk of bisphenol exposure, home-cooked meals are recommended over convenience products for children.

It seems important to provide education for parents and children on how to limit products with potential health hazards and to promote appropriate cooking and preparation techniques that minimize negative health effects. Food safety monitoring should also be increased at the production stage by incorporating good production practices. There is an immediate need for prospective studies that can evaluate cohort-based exposure to contaminants in baby food in the context of the long-term health status of children.

## Figures and Tables

**Figure 1 nutrients-13-02358-f001:**
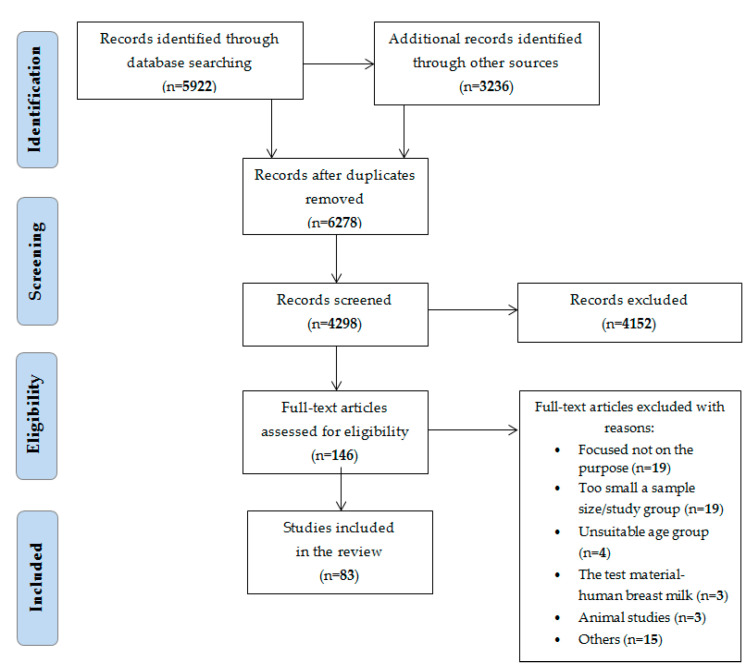
Preferred Reporting Items for Systematic Reviews and Meta-Analyses (PRISMA) flow diagram for studies retrieved through the searching and selection process with its own modifications.

**Table 1 nutrients-13-02358-t001:** Characteristics of safe levels of contaminant intake for infants and toddlers [3,4,5,6,7,8].

Type of Contamination	Safe Contamination Levels
Acrylamide	RfD = 2 μg/kg/bw
Arsenic	PTWI = 15 μg/kg/bw
Bisphenol A	TDI = 50 μg/kg/bw
Cadmium	MTD = 4.1 μg/day, PTWI = 7 μg/kg/bw
Dioxins	TDI = 4 μg/kg/bw, TWI = 2 pg/kg/bw
Furan	ADI = 2 μg/kg/bw, RfD = 1 μg/kg/bw
Lead	MTD = 0.5 μg/day, PTWI = 25 μg/kg/bw
Mercury	PTWI = 1.6 μg/kg/bw
Mycotoxins	MCL_aflatoxins_ = 50 ng/kg/bw
Nitrates, nitrites	ADI_total_ = 0–3.7mg/kg/bw
Residue pesticides	MRL = 0.1 mg/kg/bw

ADI—Acceptable Daily Intake, bw—body weight, MCL—Maximum Consented Limit, MRL—Maximum Residue Limits, MTD—Maximum Tolerated Dose, PTWI—Provisional Tolerable Weekly Intake, RfD—Reference dose, TDI—Tolerable Daily Intake, TWI—Tolerable Weekly Intake.

**Table 2 nutrients-13-02358-t002:** Summary of studies on the safety of toxic elements in baby foods.

Type of Contamination	Number of Samples	Type of Food Sample	Results	Country [Reference]
**CONTAMINANT CONTENT**				
Pb, Cd As, Hg	76	^c^ Infant formula and milk	No sample was contaminated	Norway [24]
Pb and Cd	564	^a^ Infant food	Contamination of 37% of samples with Pb and 57% with Cd, RfD exceeded by 0–3%, Pb and Cd by 6–23%	Miami, USA [10]
Pb and Cd	114	^b^ Baby desserts, juices, dinners	No sample exceeded the MCL	Poland [12]
Pb and Cd	55	^c^ Infant formula and milk	The median concentration of Pb exceeded the MCL	Brazil [13]
Pb and Cd	26	^a^ Infant food, ^b^ commercial food	Excess of Pb in 96% samples and Cd in 53% of samples	Nigeria [14]
Pb and As	90	^a^ Infant food	Pb only in 2 samples, As in 3 samples, RfD exceeding of Cd and Pb	Spain [11]
Hg	291	^a^ Infant food and ^b^ commercial food for toddlers	No sample exceeded the MCL	France [16]
Hg	150	^c^ Infant formula	No sample exceeded the standards	Canada [19]
Hg	45	^b^ Commercial food for toddlers	66% of fish samples exceeded the MRL	Menorca, Spain [17]
As	48	^b^ Commercial food for toddlers	91% of samples contained inorganic As, 100% exceeded total As	Columbia [22]
As	39	^b^ Commercial food for toddlers	Contamination of 75% of samples, exceeding the EU maximum levels	Australia [21]
Cd	18	^c^ Infant food	Cd was exceeded in rice-based products	Sweden [23]
**ESTIMATION OF INTAKE**				
Pb, Cd and Hg	119	^b^ Commercial food for toddlers	Cd and Pb concentrations exceeded TWI (35%, 42%), Hg did not exceed TWI	Korea [18]
Pb and Cd	190	^c^ Infant formula	1 sample exceeded the PTWI	Nigeria [9]
Cd and Hg	43	^b^ Commercial food for toddlers	No sample exceeded the PTWI	Tanzania [21]
Hg	87	^a^ Infant food	No sample exceeded the PTWI	Portugal [20]

As—arsenic, Cd—cadmium, Hg—mercury, MCL—Maximum Contaminant Level, MRL—Maximum Residue Limits, Pb—lead, PTDI—Provisional Tolerable Daily Intake, PTWI—Provisional Tolerable Weekly Intake, RfD—Reference dose, TWI—Tolerable Weekly Intake. ^a^ Infant food—products intended for infants up to 12 months of age as complementary foods (cereals, desserts, mousses, wafers, etc.); all solid and liquid foods other than breast milk and infant formula for infants, ^b^ Commercial food for toddlers—snacks intended for children aged 12 to 36 months, ^c^ Infant formula—food products intended for special dietary use solely for infants under 12 months of age as replacement human milk.

**Table 3 nutrients-13-02358-t003:** Summary of studies on the safety of acrylamide in foods for babies.

Type of Contamination	Number of Samples	Type of Food Sample	Results	Country [Reference]
CONTAMINANT CONTENT				
	141	^a^ Infant food, ^c^ commercial food	7% exceeded the RfD	France [26]
Acrylamide	141	^a^ Infant food, ^b^ commercial food for toddlers	Detected in 80% of samples, exceeded the RfD	France [27]
	70	^a^ Infant food, ^c^ commercial food	Exceedance of RfD in baby food	Estonia [28]
**ESTIMATION OF INTAKE**				
Acrylamide	2517	^a^ Infant food and ^c^ commercial food	Exposure of children twice as high as that of adults	USA [25]
	111	^a^ Infant food	Average intake by infants and children exceeded RfD	Poland [29]

RfD—Reference Dose, ^a^ Infant food—products intended for infants up to 12 months of age as complementary foods (cereals, desserts, mousses, wafers, etc.); all solid and liquid foods other than breast milk and infant formula for infants, ^b^ Commercial food for toddlers—snacks intended for children aged 12 to 36 months, ^c^ Commercial food—food products intended for consumption by all age groups.

**Table 4 nutrients-13-02358-t004:** Summary of studies on the safety of bisphenol A in baby foods.

Type of Contamination	Number of Samples	Type of Food Sample	Results	Country [Reference]
CONTAMINANT CONTENT				
Bisphenol A	154	^b^ Commercial food for toddlers, ^d^ commercial food	Detected in 36% of infant food samples, no TDI exceeded	China [33]
	122	^a^ Jarred infant food	Detected in 81% of samples, no MDL exceeded	Canada [30]
	103	Human milk and ^c^ infant formula	Detected in 38% of infant food samples, no RfD exceeded	Spain [31]
	76	^c^ Infant formula	None detected	China [35]
	68	^c^ Infant formula	No bisphenol exceedances, no RfD exceeded	India [34]
	50	^c^ Infant formula	Detected in 60% of infant formula samples	Italy [32]

RfD—Reference Dose, TDI—Tolerable Daily Intake, ^a^ Infant food—products intended for infants up to 12 months of age as complementary foods (cereals, desserts, mousses, wafers, etc.); all solid and liquid foods other than breast milk and infant formula for infants, ^b^ Commercial food for toddlers—snacks intended for children aged 12 to 36 months, ^c^ Infant formula—food products intended for special dietary use solely for infants under 12 months of age as replacement human milk, ^d^ Commercial food—food products intended for consumption by all age groups.

**Table 5 nutrients-13-02358-t005:** Summary of studies on the safety of dioxins in food for babies.

Type of Contamination	Number of Samples	Type of Food Sample	Results	Country [Reference]
CONTAMINANT CONTENT				
Dioxins	163	Commercial food, human milk, infant formula	No exceedance of the upper limit of the standard	Greece [40]
**ESTIMATION OF INTAKE**				
Dioxins	180	^a^ Infant food, ^b^ commercial food for toddlers	TDI exceeded in children of 7–12 months by 4.5%, in 13–36-month- olds by 5.1–7.4%	France [41]
	63	^a^ Infant food, ^b^ commercial food for toddlers	Consumption is below the TWI	Italy [42]
	60	^a^ Infant food, ^c^ infant formula	No exceedance of the TDI in baby food	Germany [39]
	16	^a^ Infant food	No exceedance of the TDI in baby food	Spain [37]

TDI—Tolerable Daily Intake, TWI—Tolerable Weekly Intake, ^a^ Infant food—products intended for infants up to 12 months of age as complementary foods (cereals, desserts, mousses, wafers, etc.); all solid and liquid foods other than breast milk and infant formula for infants, ^b^ Commercial food for toddlers—snacks intended for children aged 12 to 36 months, ^c^ Infant formula—food products intended for special dietary use solely for infants under 12 months of age as replacement human milk.

**Table 6 nutrients-13-02358-t006:** Summary of studies on the safety of furan in food for babies.

Type of Contamination	Number of Samples	Type of Food Sample	Results	Country [Reference]
CONTAMINANT CONTENT				
Furan	134	^a^ Infant food, ^b^ commercial food for toddlers	Furan found in 84% of samples	France [43]
	101	^b^ Commercial food for toddlers	Contamination of 12% of samples	Taiwan [48]
**ESTIMATION OF INTAKE**				
Furan	301	^a^ Infant food	EDI exceeded reference dose	Poland [47]
	191	^b^ Commercial food for toddlers, ^c^ commercial food	EDI about 3 times higher among infants than adults	China [45]
	78	^a^ Infant food, ^b^ commercial food for toddlers	EDI 3.8 times higher among infants than adults	Belgium [46]
		**EXPOSURE**		
	230	^a^ Infant food, ^b^ commercial food for toddlers	Medium exposure is not a health risk	Germany [49]
Furan	76	^a^ Infant food, ^b^ commercial food for toddlers	Meat- and fish-based products potential risk for children	Spain [44]

EDI—Estimated Daily Intake, ^a^ Infant food—products intended for infants up to 12 months of age as complementary foods (cereals, desserts, mousses, wafers, etc.); all solid and liquid foods other than breast milk and infant formula for infants, ^b^ Commercial food for toddlers—snacks intended for children aged 12 to 36 months, ^c^ Commercial food—food products intended for consumption by all age groups.

**Table 7 nutrients-13-02358-t007:** Summary of studies on the safety of mycotoxins in food for babies.

Type of Contamination (Number of Mycotoxins Tested)	Number of Samples	Type of Food Sample	Results	Country [Reference]
CONTAMINANT CONTENT				
Mycotoxins	1207	^c^ Infant formula, milk	Only 1% of samples exceeded the norm	China [60]
Mycotoxins (14)	215	^a^ Infant food (cereal products)	Contamination of 31% of cereals, 19% of baby cereals; norms were exceeded	Brazil [51]
Mycotoxins (1: aflatoxin M1)	185	^a^ Infant food (dairy products), ^c^ infant formula	85% of infant formula samples exceeded the MCL	Jordan [54]
Mycotoxins (5)	137	^a^ Infant food (cereal and nuts products)	Contamination of 42% of baby food samples	Nigeria [56]
Mycotoxins (1: aflatoxin M1)	101	^a^ Infant food (dairy products), ^c^ infant formula	1 sample of infant formula contaminated	Serbia [59]
Mycotoxins (1: aflatoxin M1)	84	^c^ Infant formula	Contamination of 3% of samples, norms were not exceeded	Turkey [58]
Mycotoxins (1: ochratoxin A)	64	^a^ Infant food (cereal products)	Contamination of 41% of cereal samples	Iran [55]
Mycotoxins (5)	60	^a^ Infant food (cereal products)	Contamination of 20% of cereal samples, of which 10% exceeded maximum level	Spain [52]
**ESTIMATION OF INTAKE**				
Mycotoxin (1: Deoxynivalenol)	3309	^a^ Infant food, ^c^ infant formula, milk	Average exposure twice as high as TDI	Norway [57]
Mycotoxins (1: patulin)	610	^b^ Commercial food for toddlers (apple-based products)	No exceedances of PMTDI standards	Qatar [50]
Mycotoxins (5)	302	^a^ Infant food and ^b^ commercial food for toddlers (cereal products)	No exceedances of TDI standards	Poland [53]

MCL—Maximum Contaminant Level, PMTDI—Provisional Maximum Tolerable Daily Intake, TDI—Tolerable Daily Intake, ^a^ Infant food—products intended for infants up to 12 months of age as complementary foods (cereals, desserts, mousses, wafers, etc.); all solid and liquid foods other than breast milk and infant formula for infants, ^b^ Commercial food for toddlers—snacks intended for children aged 12 to 36 months, ^c^ Infant formula—food products intended for special dietary use solely for infants under 12 months of age as replacement human milk.

**Table 8 nutrients-13-02358-t008:** Summary of studies on the safety of nitrates and nitrites in food for babies.

Type of Contamination	Number of Samples	Type of Food Sample	Results	Country [Reference]
CONTAMINANT CONTENT				
Nitrites, nitrates	1319	^b^ Commercial food for toddlers, ^c^ commercial food	ADI of nitrite was exceeded in 16% of infant and 58% of toddler food samples	France [67]
Nitrites, nitrates, N-nitrosoamines	315	^b^ Commercial food for toddlers (meat)	ADI of nitrite was exceeded in 40% and 29% of child food samples at different times	Estonia [68]
Nitrites, nitrates	157	^b^ Commercial food for toddlers (meat)	ADI of nitrite was exceeded in 3% of child food samples	Estonia [66]
Nitrates, nitrites	108	^a^ Infant food, ^c^ commercial food (vegetable, fruit, cereals and milk based)	Average nitrite content in the upper limit of the standard	Fiji [65]
**ESTIMATION OF INTAKE**				
Nitrates	1150	^b^ Commercial food for toddlers (vegetable based), vegetables	No exceedance of the maximum permissible dose	Spain [62]
Nitrites, nitrates	104	^a^ Infant food, ^c^ commercial food	No exceedance of the maximum permissible dose	Italy [63]
Nitrates	80	^a^ Infant food (vegetable based)	Only 1 sample exceeded the ADI	Portugal [61]
Nitrates	39	^a^ Infant food (vegetable, meat- based)	No exceedance of the maximum permissible dose	Portugal [64]

ADI—Acceptable Daily Intake, ^a^ Infant food—products intended for infants up to 12 months of age as complementary foods (cereals, desserts, mousses, wafers, etc.); all solid and liquid foods other than breast milk and infant formula for infants, ^b^ Commercial food for toddlers—snacks intended for children aged 12 to 36 months, ^c^ Commercial food—food products intended for consumption by all age groups.

**Table 9 nutrients-13-02358-t009:** Summary of studies on the safety of pesticide residues in food for babies.

Type of Contamination (Number of Pesticides Tested)	Number of Samples	Type of Food Sample	Results	Country [Reference]
CONTAMINANT CONTENT				
Pesticides (86)	522	^a^ Infant food (fruit based)	Detected in 60% of samples, 1.4% exceeded the MRL	Czech Republic [70]
Pesticides (516)	309	^b^ Commercial food for toddlers, ^c^ commercial food	Detected in 67% of samples, exceeded the TRV	France [69]
Pesticides: (1: imidacloprid)	250	^a^ Infant food, fruit, vegetables, cereal	Detected in 15% of samples, exceeded the MRL	India [72]
Pesticides (4: glyphosate, glufosinate, perchlorate, chlorate)	105	^a^ Infant food, ^c^ commercial food	Perchlorate detected in 10.5% of samples	Italy [75]
Pesticides (18)	100	^c^ Commercial food (homemade)	Detected in 100% of samples	Korea [71]
Pesticides (69)	54	^a^ Infant food (juices, purees)	Detected in 56% of samples	Serbia [72]
Pesticides (12)	33	^a^ Infant food (juices, multi-fruit jars)	Three pesticides detected in 60% of samples	Spain and United Kingdom [74]

TRV—toxicological reference value, ^a^ Infant food—products intended for infants up to 12 months of age as complementary foods (cereals, desserts, mousses, wafers, etc.); all solid and liquid foods other than breast milk and infant formula for infants, ^b^ Commercial food for toddlers—snacks intended for children aged 12 to 36 months, **^c^** Commercial food—food products intended for consumption by all age groups.

**Table 10 nutrients-13-02358-t010:** Summary of studies on the safety of PAH in food for babies.

Type of Contamination	Number of Samples	Type of food Samples	Results	Country [Reference]
**CONTAMINANT CONTENT**				
PAH	126	^b^ Commercial food (meat, fish)	No exceedances in meat and fish, exceedances in dairy products	Italy [77]
	40	^a^ Infant food, ^b^ commercial food (dairy products)	Exceedance in dairy samples 18.2%, meat and fish 5.6%	Italy [76]
**ESTIMATION OF INTAKE**				
PAH	322	^b^ Commercial food (meat)	No exceedances in average PAH intake	Estonia [78]
	42	^a^ Infant food	One sample exceeding MCL	Iran [79]
**EXPOSURE**				
PAH	152	^c^ Infant formula	No exceedances (MOE > 10,000)	Korea [81]
	40	^c^ Infant formula	No exceedances (MOE > 10,000)	Nigeria [80]

MCL—Maximum Consumed Limit, MOE—Margin of Exposure, PAH—polycyclic aromatic hydrocarbons, BaP-benzopyrene, ΣPAH4, sum of subgroup of four hydrocarbons, ^a^ Infant food—products intended for infants up to 12 months of age as complementary foods (cereals, desserts, mousses, wafers, etc.); all solid and liquid foods other than breast milk and infant formula for infants, ^b^ Commercial food—food products intended for consumption by all age groups, ^c^ Infant formula—food products which are represented for special dietary use solely for infants under 12 months of age as replacement human milk.

**Table 11 nutrients-13-02358-t011:** Summary of studies on the safety of 3-MCPD and glycidyl esters in food for babies.

Type of Contamination	Number of Samples	Type of Food Sample	Results	Country [Reference]
CONTAMINANT CONTENT				
Glycidyl esters, 3-MCPD	275	^a^ Infant food (homemade), ^c^ commercial food for toddlers	Over 71% of diet samples contaminated with 3-MCPD and glycidyl esters	China [85]
Glycidyl esters, 3-MCPD	222	^a^ Infant food	Lower concentrations of 3-MCPD (7-fold) and glycidyl esters (3-fold) during 3 years	USA [82]
Glycidyl esters, 3-MCPD	130	^a^ Infant food, ^c^ commercial food	Products containing palm oil had a higher concentration of 3-MCPD and glycidyl esters	Italy [84]
Glycidyl esters, 3-MCPD	96	^a^ Infant food	Lowest concentrations in products containing palm oil	USA [87]
Glycidyl esters, 2-MCPD, 3-MCPD	77	^a^ Infant food	2-MCPD detected in all samples	Germany [83]
**EXPOSURE**				
3-MCPD	60	^c^ Commercial food for toddlers	Exceeded TDIs in potato chips, crackers, peanuts, muesli, and biscuits	Poland [86]

3-MCPD—3-monochloropropane-1,2-diol, 2-MCPD— 2-monochloropropane-1,2-diol, TDI—the tolerable daily intake, ^a^ Infant food—products intended for infants up to 12 months of age as complementary foods (cereals, desserts, mousses, wafers, etc.); all solid and liquid foods other than breast milk and infant formula for infants, ^c^ Infant formula—food products which are represented for special dietary use solely for infants under 12 months of age as replacement human milk.

**Table 12 nutrients-13-02358-t012:** Summary of studies on the safety of MOH in food for babies.

Type of Contamination	Number of Samples	Type of Food Sample	Results	Country [Reference]
CONTAMINANT CONTENT				
MOH	51	^c^ Infant formula	MOH detected in 33%	China [88]
	50	^c^ Infant formula	MOH detected in 66%	China [89]
	16	^a^ Infant food	MOH detected in all samples containing meat and fish	Italy [8]
**EXPOSURE**				
MOH	230	^a^ Infant food, ^b^ commercial food for toddlers, ^c^ infant formula	Highest risk of MOH exposure among infants	China [90]

MOH—mineral oil hydrocarbons, ^a^ Infant food—products intended for infants up to 12 months of age as complementary foods (cereals, desserts, mousses, wafers, etc.); all solid and liquid foods other than breast milk and infant formula for infants, ^b^ Commercial food for toddlers—snacks intended for children aged 12 to 36 months, ^c^ Infant formula—food products which intended for special dietary use solely for infants under 12 months of age as replacement human milk.

## Data Availability

Detailed data are available from the authors.

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
