# Peer review of "Assessment of the Risk of Contamination of Food for Infants and Toddlers"

_nutrients, 2021, doi:10.3390/nu13072358_

Round 1

Reviewer 1 Report

This is a big review in a broad research area focusing on an important vulnerable group. Tables 3-13 collect important contaminant groups. The studies in the tables are reported in an order from higher to lower number of samples/participants, which is one possible logic. Is it consistently mentioned in the tables, if reference values are exceeded? e.g. in Table 3, it is said that 37% of the infant food samples are contaminated with lead (US, ref. 86), does this mean that the values did not exceed the reference values as that was not mentioned?

In Table 9, why are the ADI values not presented separately for nitrate and nitrite? 

As the Nordic countries (Sweden, Denmark, Iceland, Norway, Finland) are following carefully the content in foods and the intakes of many contaminants also in children, I am wondering why there are no studies in the Tables from these countries?

There are very large differences between regions and countries in the contents and intakes of many contaminants. The mycotoxins in cereals are partly different and their content also differ between Northern and Southern Europe, for instance.  These geographical differences should be discussed.

Reviewer 2 Report

Few correction needed. Find out the file.  

Author Response

This manuscript is a resubmission of an earlier submission. The following is a list of the peer review reports and author responses from that submission.

Round 1

Reviewer 1 Report

Dear Authors,

the current review describes the contaminants found in you infants and young children foods, the review used an extended list of bibliographic information and brings together different type of contaminants. Nevertheless, the information given is provided in several other manuscripts, being the only novelty the way that authros presented in this work, but the information provided, per se is not novel.

The manuscript needs minor revision, for revision of minor English errors

Reviewer 2 Report

-Why is are groups “food for infants and young children” used in title, even though this manuscript only concerns baby foods and infant formulas (used search terms) ?

- Section Materials and methods is very short and does n information, not give all the necessary information, e.g. which database(s) were used in data retrieval.

- Introduction is quite long.  Instead of writing a paragraph for each contaminant group, a table that reviews sources and effects in table format (e.g. extension of Table 1. in horizontal position). could be easier for readers and similar effects caused by different toxins could be more prominent.

- How were included contaminants or contaminant groups selected? Outside were left significant contaminants such as glycidol esters and mineral oil hydrocarbons.

- In section Results and conclusions contaminant content, intake, and exposure are addressed without separation. Consequently, studies and results in tables are not comparable.

- No definition of food groups (e.g Food products for infants and toddlers, Commercial food, food products for children) in tables in section Results and conclusions. This makes comparison between groups and studies difficult.

- Why are pesticides included to this review even though they are a very large and special group (in terms of  exposure) of substances with wide variety of adverse effects ?

- Acrylamide: “infant snacks” , potato chips? For which age groups are these relevant? These are not baby foods or infant formulas.

- Row 277: potulin?

Round 2

Reviewer 2 Report

Modifications of the manuscript did not significantly improve its quality.